# Population Substructure Has Implications in Validating Next-Generation Cancer Genomics Studies with TCGA

**DOI:** 10.3390/ijms20051192

**Published:** 2019-03-08

**Authors:** Marina D. Miller, Eric J. Devor, Erin A. Salinas, Andreea M. Newtson, Michael J. Goodheart, Kimberly K. Leslie, Jesus Gonzalez-Bosquet

**Affiliations:** 1Department of Obstetrics and Gynecology, University of Iowa Hospitals and Clinics, Iowa City, IA 52242, USA; marina-miller@uiowa.edu (M.D.M.); eric-devor@uiowa.edu (E.J.D.); kimberly-leslie@uiowa.edu (K.K.L.); 2Holden Comprehensive Cancer Center, University of Iowa Hospitals and Clinics, Iowa City, IA 52242, USA; michael-goodheart@uiowa.edu; 3Compass Oncology, Portland, OR 97227, USA; Erin.Salinas@compassoncology.com; 4Division of Gynecologic Oncology, Department of Obstetrics and Gynecologic, University of Iowa Hospitals and Clinics, Iowa City, IA 52242, USA; andreea-newtson@uiowa.edu

**Keywords:** population substructure, genetic admixture, endometrial cancer, ovarian cancer, The Cancer Genome Atlas

## Abstract

In the era of large genetic and genomic datasets, it has become crucially important to validate results of individual studies using data from publicly available sources, such as The Cancer Genome Atlas (TCGA). However, how generalizable are results from either an independent or a large public dataset to the remainder of the population? The study presented here aims to answer that question. Utilizing next generation sequencing data from endometrial and ovarian cancer patients from both the University of Iowa and TCGA, genomic admixture of each population was analyzed using STRUCTURE and ADMIXTURE software. In our independent data set, one subpopulation was identified, whereas in TCGA 4–6 subpopulations were identified. Data presented here demonstrate how different the genetic substructures of the TCGA and University of Iowa populations are. Validation of genomic studies between two different population samples must be aware of, account for and be corrected for background genetic substructure.

## 1. Introduction

The Cancer Genome Atlas (TCGA) was launched in 2008 as a collaboration between the National Institutes of Health (NIH) and the National Human Genome Research Institute (NHGRI) with an initial aim to catalogue genomic alterations of three cancers: glioblastoma multiforme, lung, and ovarian cancer. Shortly thereafter, the project expanded to profiling 30 cancer genomes. This initiative opened doors for a wide variety of research efforts with the long-term goal of improving diagnosis, treatment, and cancer prevention [1]. Thousands of publications have been made possible using data derived from TCGA. One of the many advantages of having such a large set of publicly available data is study validation. Prior to the introduction of such large databases, in order to validate findings from a study, the study would have had to be repeated in an independent dataset. However, with the availability of genomic data from TCGA, validating findings of genomic studies has become much more readily available and less labor intensive than repeating the study in another institution. 

Although TCGA and other publicly available datasets have made these validations much simpler, the limitations of doing so have yet to be fully addressed. For example, in our previous studies designed to predict clinical outcomes by integrating clinical, pathological and molecular features of patients with cancer, we found that the best prediction models, developed using our internal patient cohort (University of Iowa), performed 10–20 percentage points worse in TCGA datasets [2,3]. Based on the characteristics of our population (northern European origin), it was obvious that our patients’ backgrounds seemed to be more homogeneous than the population from which TCGA derived their datasets. Furthermore, emerging studies in breast cancer have detected differences in gene expression and clinical outcomes based in ethnicity, even after accounting for other clinical and social factors [4]. Similar effects of ethnicity in cancer outcomes have been reported in endometrial cancer, despite accounting for other clinical and social factors [5,6]. Thus, we started to question the generalizability of TCGA data as a validation platform for populations with dissimilar compositions. 

Therefore, we hypothesized that populations in different regions of the United States and the world that are ethnically different are likely genetically different than the population from which TCGA samples were derived. This has the potential to limit the power of TCGA data to validate data from independent datasets. The goal of the present study is to determine the genetic composition derived from sequencing ovarian and endometrial cancer samples of a patient population from our institution and compare it to the genetic composition derived from sequencing the same tumor types in TCGA population.

## 2. Results

ADMIXTURE analysis of the University of Iowa Hospitals and Clinics (UIHC) population revealed the lowest cross-validation errors with a K of 1. STRUCTURE analysis with the ΔK method does not show any structure and this is confirmed by the highest Ln probability with a K of 1 and decreasing with higher K values. The results for the UIHC analyses are summarized in Figure 1. 

ADMIXTURE analysis of the TCGA population revealed the lowest cross-validation error with a K of 4, and STRUCTURE analysis with the ΔK method reveals an optimal K of 6. When stratifying TCGA data to account for the different origins of cancer, ADMIXTURE analysis of endometrioid endometrial cancer patients revealed an optimal K of 3, and STRUCTURE analysis revealed an optimal K of 2. For high grade serous ovarian cancer, ADMIXTURE analysis revealed an optimal K of 2, and STRUCTURE analysis revealed an optimal K of 4. These data are summarized in Figure 2 and Figure 3. In Appendix A we detailed the membership coefficient matrix, termed the individual *Q-matrix*, for each analysis of TCGA cohort. The Q-matrix is generated with rows for the number of individuals analyzed and columns for K clusters. All coefficients in a row sum to 1. Because UIHC has one K cluster, the Q-matrix only would have a column with all “1” and we did not include them. 

The mean *F_ST_* for our comparison between both UIHC and TCGA cohorts was 0.015. Considering that the potential range of *F_ST_* statistics goes from 0 (for genetically similar populations) to 1 (for divergent populations), we interpreted that there were some differences between both populations. However, is difficult to make further inferences from these data (Appendix A has more details of this analysis).

## 3. Discussion

The goal of our study was not to perform a comprehensive subpopulation structure analysis of all our patients. Rather, our aim was to assess the genetic background of patients with endometrial and ovarian cancer that we diagnose and treat, and compare them with the genetic background of patients with the same cancer types in TCGA. The motivation of this study stemmed from the need to validate a prediction model of clinical outcomes that integrated clinical, pathological, and diverse molecular data: gene and miRNA expression, gene copy number and somatic mutations [2,3]. The validation of these prediction models in TCGA datasets was not ideal, so we wanted to investigate possible reasons for this discordance. We chose to study population substructure because it seemed that UIHC patients came from a more homogeneous population than TCGA patients, who were more diverse. The final population admixture analysis confirmed our hypothesis that the genetic backgrounds of both cohorts of patients were not similar. Therefore, validation of genetic studies of UIHC patients within TCGA dataset may be limited. 

These results demonstrate differences in the genetic composition of Iowa patients with ovarian and endometrial cancer from the TCGA patients with similar cancers. Given that the general population of the state of Iowa is very homogeneous and composed primarily of Caucasians (>90%) according to the United States Census Bureau these results are not surprising [7]. The TCGA cohort, on the other hand, was derived from patients receiving treatment at multiple institutions across the United States. A racial/ethnic breakdown of the TCGA showed that the uterine cancer sample was nearly 20% African American while the ovarian cancer sample was only 6% African American [8]. Further, the proportion of both sample populations that is Hispanic is 2% to 3% which is far below the 17% to 20% figure that is generally reported for the US population. These differences in genomic admixture should not be ignored as they may impact the generalizability of the findings of a TCGA study to other populations. It may also limit one’s ability to use data derived from TCGA to validate results from an independent data set. Further, the issue of racial disparities and how it affects cancer is one that merits attention. 

Cote et al wrote about the racial disparity in endometrial cancer [9]. Basing their analysis on SEER data that reports for four racial/ethnic groups: Non-Hispanic White (NHW), Non-Hispanic Black (NHB), Hispanic and Asian, there is a clear disparity in both rate of increase in diagnosis and in histology/aggressiveness among these groups. NHB have a much greater proportion of their cancers being either serous histology or malignant mixed mullerian tumors versus the other groups. NHB also have a much higher proportion of higher grade tumors and, not surprisingly, significantly lower survival rates. The Cote et al paper cites numerous other studies to back up their analyses. One of these cited studies showed that deaths from endometrial cancer actually exceeded deaths from ovarian cancer among NHB women [10]. They also observed that the experience of NHB women for endometrial cancer extends to breast cancer as well with significantly higher mortality and lower 5-year survival rates as compared with NHW women. In ovarian cancer, NHB women routinely experience poorer 5-year survival from ovarian cancer compared with other racial/ethnic groups [6]. Recently, racial/ethnic disparities in endometrial cancer were extended to include molecular differences leading to the suggestion that such differences could present therapeutic opportunities that are racial/ethnic group-specific [5]. However, this only becomes useful if the patient population composition is well defined.

So, higher incidence, more aggressive histologies and lower survival rates among NHB women translates into the very relevant question: What is the racial/ethnic composition of your endometrial cancer sample? In the state of Iowa the African American population is estimated to be 2.68% while states like Maryland and Mississippi it exceeds 30%. Similarly, the Hispanic population of Iowa is estimated to be around 6% while California, New Mexico and Texas all exceed 35%. Moreover, the term Hispanic/Latino is a loosely used amalgam. Historically, there is a substantial Native American contribution to the Hispanic populations of the American Southwest while those whose ancestors originated in the Caribbean or some Central American countries have a substantial African contribution [11].

In the present study we have shown that there are differences in the genetic background between patients with cancer in the UIHC and TCGA cohorts. What are some of the strategies that we could use to account for these differences and make our study more generalizable? In general, we could use the same strategies that have been used in genome-wide association studies (GWAS) to address this same problem [12]. First, we could stratify the validation analysis by subpopulation. In our case, we would validate our prediction model, built with a majority of patients with northern European ancestry, with TCGA patients of similar background. This approach may have limitations depending in the sample size of both cohorts to be used. Another strategy would be to adjust our expected validation threshold based on previous results of differences in prediction model performance. For example, we would expect that a prediction model built using UIHC data would have a 10–20 percentage point lower performance than validated models using TCGA data. A third strategy would be to create a correction factor that accounts for the variation of the genetic background of each subpopulation and apply this correction factor to each of the individuals introduced in the model. Principal component analysis (PCA) would be a type of correction factor that could be applied. We could co-opt these GWAS strategies for our purposes because the prediction analysis is, basically, also an association study of multiple variables with an outcome. 

Our study is limited by the retrospective nature of its design. Biases may have influenced patient selection and recruitment. However, as noted previously, the population sampled and analyzed in our study represents the general population of the State of Iowa. Furthermore, there is no definitive answer in how to choose the optimal number of clusters. The choice of an appropriate value for K is a notoriously difficult statistical problem and is somehow subjective [13]. It must be informed by the knowledge of the underlying population history. In our study we used one approximation for each method used, STRUCTURE and ADMIXTURE. Because the estimation of the K subpopulation analysis is subjected to interpretation, we cannot conclude that the differences in TCGA results are significant.

## 4. Materials and Methods 

The goal of our study was to assess whether the genetic composition of UIHC patients was different from TCGA patients. This study stemmed from the observation that our prediction models of clinical outcomes in cancer were not validating completely in TCGA data. Also, we observed that our population was predominantly from northern European descent in comparison to TCGA population. We wanted to perform the genetic admixture comparison with the available molecular material already processed, sequenced RNA, without having to process and sequence more samples. We thought that this same problem may present to other researchers and our methods may therefore be useful for their research. Herein, we describe a sequence of procedures that will extract genotypes from sequenced RNA and perform subpopulation structure analysis. In order to compare the UIHC data to TCGA data, we performed the same analyses in files resulting from RNA sequencing (BAM files) from both cohorts. 

### 4.1. Tissue Procurement

A primary tumor cohort consisting of 112 patients diagnosed and treated for endometrial or ovarian cancers at the University of Iowa Hospitals and Clinics (UIHC), retained under informed consent (IRB# 200910784 and 200209010), was assembled from the Gynecologic Tumor Bank of the Department of Obstetrics & Gynecology Women’s Health Tissue Repository [9]. Sixty-two of these patients were diagnosed with endometrioid endometrial cancer and 50 were diagnosed with high grade serous ovarian cancer. A summary of clinical characteristics, including self-declared race, are displayed in Table 1. 

### 4.2. RNA Purification and Sequencing 

Total cellular RNA was purified from individual tumors using the mirVANA mRNA isolation kit following manufacturer’s recommendations (Thermo Fisher Scientific, Waltham, MA, USA). RNA concentration and purity of the collected RNAs were assessed using a Nanodrop 1000 spectrophotometer (Thermo Fisher Scientific, Waltham, MA, USA) and a Model 2100 Bioanalyzer (Agilent, Santa Clara, CA, USA). RNA preparations with sufficient mass and integrity (RIN >7.0) were submitted to the Genomics Division of the University of Iowa Institute for Human Genetics for RNA sequencing (RNAseq) [10]. Total cellular RNA (500 ng) was fragmented, converted to cDNA and ligated to sequencing adaptors containing indexes using the Illumina TruSeq stranded total RNA library preparation kit (Illumina, Inc., San Diego, CA, USA). Molar concentrations of the indexed libraries were measured on the Model 2100 Agilent Bioanalyzer and combined equally into pools for sequencing (Agilent, Santa Clara, CA, USA). The concentration of the pools were measured using the Illumina Library Quantification Kit (KAPA Biosystems, Wilmington, MA, USA) and sequenced on the Illumina HiSeq 4000 genome sequencer using a 150 bp paired-end SBS chemistry. 

### 4.3. TCGA Cohort

Endometrial and ovarian cancer data were downloaded from The Cancer Genome Atlas (TCGA) from the National Cancer Institute, following TCGA Human Subject Protection and Data Access Policies. Molecular data from RNAseq were obtained for 395 samples of endometrioid endometrial cancer and 351 samples of high grade serous ovarian cancer (Table 1). 

### 4.4. File Pre-Processing

BAM files were obtained from RNAseq alignments and converted to VCF files for genotype extraction [14,15]. PLINK software was then implemented to filter the VCF files by minor allele frequency at *q* > 0.05 and by linkage disequilibrium (LD) < 0.1 r^2^ for pairs of markers inside 200 kb [16,17]. The filtering process was performed to identify independent markers within the sequenced samples and resulted in 39,900 markers in the UIHC cohort and in 15,599 in TCGA cohort. Then, we imputed missing genotypes for all samples utilizing BEAGLE 4.1 to obtain genotype coverage of >99% for all markers [18]. 96% of all genotypes were imputed for the UIHC cohort and 97% for TCGA cohort. After imputation 109 markers were dropped from the UIHC dataset, 15 markers dropped from TCGA cohort, and there were no missing genotypes in either UIHC or TCGA cohorts.

### 4.5. Data Analysis

For the subpopulation analysis we used ADMIXTURE and STRUCTURE software packages. Both ADMIXTURE and STRUCTURE are programs that perform model-based estimation of ancestry, or population structure, using large genotype datasets from unrelated individuals [19,20]. They are model-based methods because they assume a model in which there are K populations, while K may be unknown. Each of these K populations is characterized by a set of allele frequencies at each locus. We selected these well-known algorithms not only because they are widely used, but because they model the probability of the observed genotypes using ancestry proportions and population allele frequencies. Ancestry proportions may be used to account for subpopulation structure in validation of prediction models. Other types of approaches based on algorithmic ancestry estimation use multivariate analysis techniques, like cluster analysis and principal component analysis (PCA), and do not provide individual ancestry proportions [19].

STRUCTURE takes a Bayesian approach and relies on a Markov chain Monte Carlo (MCMC) algorithm to sample the posterior distribution [20]. ADMIXTURE uses the same likelihood model but focuses on maximizing the likelihood rather than on sampling the posterior distribution. ADMIXTURE runs faster than STRUCTURE due to a fast block relaxation scheme [19]. Utilizing STRUCTURE package, subpopulation analysis was performed assuming varying numbers of clusters, or subpopulations (K) from 1 to 15 and inferring the best K using the Evanno K method [20,21]. Next, genomic admixture was determined utilizing the ADMIXTURE package. The “best fit” model was then determined based on the K that exhibited a low cross-validation error compared to other values. The best model for both STRUCTURE and ADMIXTURE methods was used to determine the percentage of each subpopulation in each sample and then all of them were reported in bar plot representation [22]. The same analysis was carried out for both the UIHC and TCGA cohorts. 

Also, as measure of genetic variation in different populations, and to compare to the other methods, we determined the fixation index (or *F_ST_*) [23]. *F_ST_* statistic can be estimated from genetic polymorphism data, such as single-nucleotide polymorphisms (SNPs) or microsatellites. However, interpretation of results may be difficult with highly variable markers and different population sizes [24]. For this analysis we used the R package *SNPRelate* [25].

## 5. Conclusions

Understanding patient population substructure is important to better understand their disease process and to lend context to differences seen therein. Further, gaining knowledge of the composition of the study population is essential to sensibly stratifying the results. The implications of genetic heterogeneity at the subpopulation level are noteworthy, including the potential for sub-population stratification in therapeutic interventions.

## Figures and Tables

**Figure 1 ijms-20-01192-f001:**
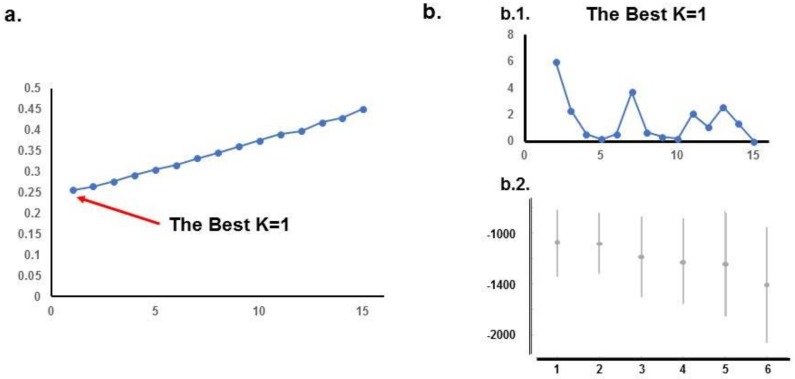
STRUCTURE and ADMIXTURE subpopulations structure analysis of UIHC patients. (**a**). ADMIXTURE analysis: Cross-validation error is minimal when K = 1; (**b**). STRUCTURE analysis: b.1. K method does not show any structure; b.2. Ln probability is higher for K = 1 and decreases for higher K values; both these results support the idea that UI sample of the population has no structure.

**Figure 2 ijms-20-01192-f002:**
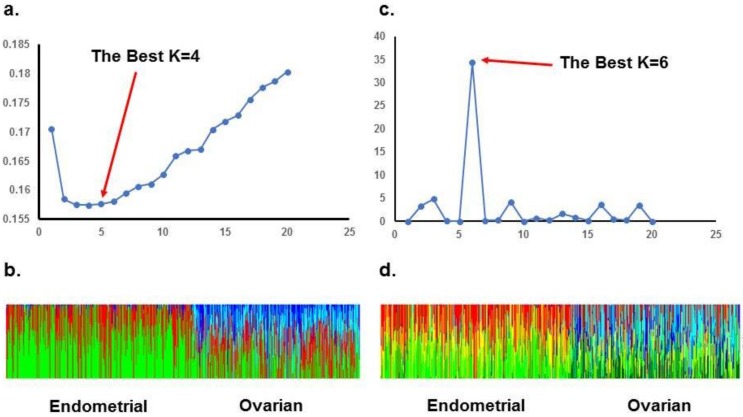
STRUCTURE and ADMIXTURE subpopulations structure analysis of TCGA patients: (**a**). ADMIXTURE analysis: Cross-validation error is minimal when K = 4; (**b**). Bar plot of admixture results for admixture proportions organized by origin of tumor; (**c**). STRUCTURE analysis: K methods shows a best K = 6; (**d**). Bar plot of STRUCTURE results for admixture proportions organized by origin of tumor.

**Figure 3 ijms-20-01192-f003:**
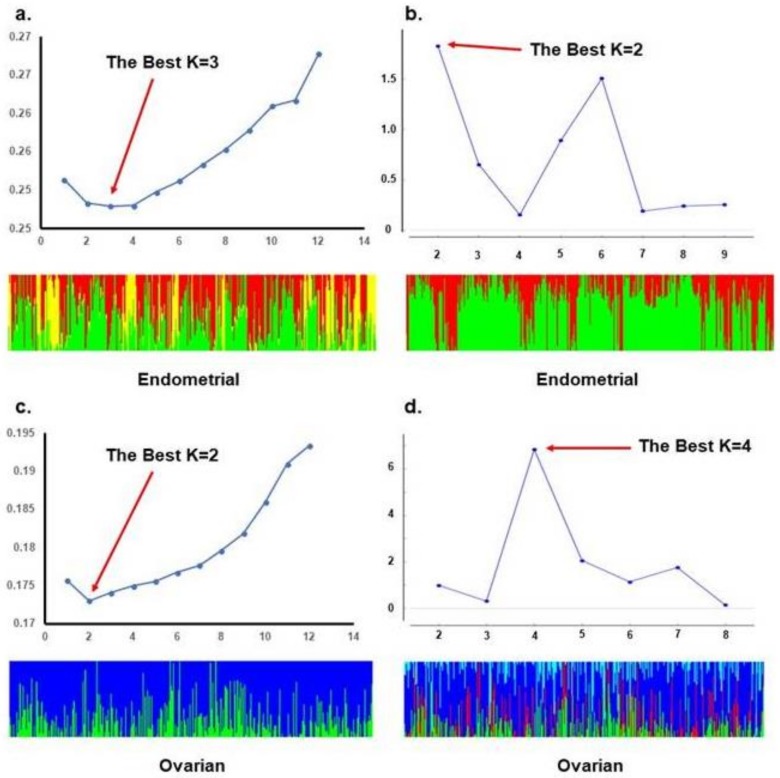
STRUCTURE and ADMIXTURE subpopulations structure analysis of TCGA patients based on the origin of cancer: (**a**). ADMIXTURE analysis and bar plot with an optimal K = 3 subpopulation substructure for endometrial cancer patients of endometrioid type; (**b**). STRUCTURE analysis and bar plot with an optimal K = 2 subpopulation substructure for endometrial cancer patients of endometrioid type; (**c**). ADMIXTURE analysis and bar plot with an optimal K = 2 subpopulation substructure for serous ovarian cancer patients; (**d**). STRUCTURE analysis and bar plot with an optimal K = 4 subpopulation substructure for serous ovarian cancer patients.

**Table 1 ijms-20-01192-t001:** Patient clinical characteristics. Data is divided by tumor type and by origin of samples (University of Iowa, UIHC, or TCGA). * Self-reported race and ethnicity.

	UIHC		TCGA	
**Cancer**	Ovarian	Endometrial	Ovarian	Endometrial
**Histological Type**	High grade serous	Endometrioid	High grade serous	Endometrioid
**Samples**	50	62	351	395
**Age (mean)**	59	61	59	65
*** Race:**				
**White**	48	57	302	288
**Black**	1	0	25	61
**Asian**	0	0	10	17
**Pacific Islander**	0	1	1	7
**American Indian**	0	0	2	3
**Unknown**	1	4	12	20
*** Ethnicity**				
**Hispanic**	0	0	8	9
**Non-Hispanic**	49	58	201	275
**Unknown**	1	4	142	111
**Stage:**				
**I**	0	44	1	281
**II**	0	4	20	34
**III**	34	11	274	66
**IV**	13	3	53	14
**Unknown**	3	0	1	1

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
