# Peer review of "Population Substructure Has Implications in Validating Next-Generation Cancer Genomics Studies with TCGA"

_ijms, 2019, doi:10.3390/ijms20051192_

Round 1

Reviewer 1 Report

The objective of this study was to evaluate the extent to which TCGA is a relevant validation set for independent studies based on a comparison of subpopulation structure and genetic admixture of an ovarian and endometrial cancer patient cohort from the University of Iowa to their counterparts in TCGA. The manuscript is well written. However, I have several questions regarding the motivation, rationale, and interpretation of these analyses, which I have listed below.

Admixture was unlikely to be present in the Iowa patient cohort because they are all Caucasian, and indeed the authors found the optimal K=1 population substructure. The rationale for testing this was unclear. It seems as though there were bound to be differences when comparing to a more heterogenous population such as TCGA. Also, wouldn’t there be advantages to validating findings from a homogenous population in a more heterogeneous population? 

The authors observe differences between these two patient cohorts but do not venture beyond reporting that the differences exist. I would be interested in a more thorough evaluation and discussion of potential solutions to the issues posed regarding generalizability. What are strategies that can be used to either (i) increase generalizability of independent studies, (ii) select an appropriate validation cohort, and (iii) account for population substructure?

The authors state that validating genetic studies with TCGA is limited because the genetic composition of the TCGA patients may be different that the patients in the independent study cohort. However, wouldn’t population substructure primarily impact the results from a GWAS if the significant SNPs varied in frequency between populations? 

Have the authors considered how Fst values may be used in this context?

Why did the authors choose ADMIXTURE and STRUCTURE? Given this is presented as a methods paper, greater detail about these algorithms and how they differ from each other would be appreciated. Is there significance to the differences in optimal K subpopulation structure for TCGA patients detected between the two algorithms? 

The central issue of accounting for population substructure in a validation study is relevant beyond TCGA. Why did the authors frame it as an issue specific to TCGA?

Minor: the authors should consider changing the spelling of “publically” to the more commonly used “publicly”

Author Response

Reviewer 1.

The objective of this study was to evaluate the extent to which TCGA is a relevant validation set for independent studies based on a comparison of subpopulation structure and genetic admixture of an ovarian and endometrial cancer patient cohort from the University of Iowa to their counterparts in TCGA. The manuscript is well written. However, I have several questions regarding the motivation, rationale, and interpretation of these analyses, which I have listed below.

1. Admixture was unlikely to be present in the Iowa patient cohort because they are all Caucasian, and indeed the authors found the optimal K=1 population substructure. The rationale for testing this was unclear. It seems as though there were bound to be differences when comparing to a more heterogenous population such as TCGA. Also, wouldn’t there be advantages to validating findings from a homogenous population in a more heterogeneous population? 

The reviewer is exactly on point. The motivation for this study (and manuscript) arose from our research in outcomes prediction for endometrial cancer (both studies are in varying stages of peer review). To validate our results, we needed a comprehensive molecular database in endometrial cancer that also had clinical information; the obvious selection was The Cancer Genome Atlas (TCGA). Validation of our prediction models in TCGA was not as accurate as we would have anticipated. Thus, we started to investigate potential reasons. Looking at the characteristics of our population, it was obvious that our patients’ backgrounds seemed to be more homogeneous (northern European) than the population used in TCGA. However, we wanted to quantify the difference between both samples of these patients with gynecologic cancer in order to find a way to account for these differences.

In this revised manuscript, we have added a sentence in the Introduction section to explain in better detail our motivation for the present study. Also, we added a sentence in the first paragraph of the Discussion section to highlight the take home message for further studies and to emphasized how important it is to understand the population from which the cases are extracted. Also, we added new 2 references to support our motivation (Salinas, E. A.; Miller, M. D.; Newtson, A. M.; Sharma, D.; McDonald, M. E.; M.E., K.; Smith, B. J.; Bender, B. J.; Goodheart, M. J.; Thiel, K. W.; Devor, E. J.; Leslie, K. K.; Gonzalez-Bosquet, J., A prediction model for preoperative risk assessment in endometrial cancer utilizing clinical and molecular variables. International Journal of Molecular Sciences. 2019 (under review); Miller, M. D.; Salinas, E. A.; Newtson, A. M.; Sharma, D.; M.E., K.; Warrier, A.; Smith, B. J.; Bender, B. J.; Goodheart, M. J.; Thiel, K. W.; Devor, E. J.; Leslie, K. K.; Gonzalez-Bosquet, J., An Integrated Prediction Model of Recurrence in Endometrial Endometrioid Cancers. Cancer Management and Research. 2019 (under review)).

2. The authors observe differences between these two patient cohorts but do not venture beyond reporting that the differences exist. I would be interested in a more thorough evaluation and discussion of potential solutions to the issues posed regarding generalizability. What are strategies that can be used to either (i) increase generalizability of independent studies, (ii) select an appropriate validation cohort, and (iii) account for population substructure?

We agree with the reviewer that this would be an interesting part of the Discussion, although at this point it is a bit speculative.  However, the results of our study pointed out to several options for validation of prediction models in populations that are not quite homogeneous:

                1. Stratified analysis: The most direct method would be to use only patients of the same background for validation. This option may be limited by the sample size: if we have to decrease the number of patients in both the testing set and validation set to make them more homogeneous from the population substructure point of view, the accuracy of the prediction model may decrease.

                2. Threshold adjustment: If sample size makes that difficult or non-feasible, then we would have to take into account a performance detriment of 10-20 percentage points when comparing results from a homogenous population with a more heterogeneous dataset.

                3. Correction of stratification: Another possibility would be to create a correction factor that would account for different backgrounds or population substructure. This method has been used in GWAS analysis to correct for stratification via principal components analysis (PCA) and other methods. For example, we would validate the prediction model for each of the different subpopulations in TCGA. Next, for each population, we will generate a correction factor specific for that population that explains the variability due to genetic background. This factor will then be applied to each patient when we perform the validation studies in the full TCGA dataset.

We added a new paragraph in the Discussion section to outline these possible methods to account for population stratification. Also, we added a new reference: Sillanpaa, M. J., Overview of techniques to account for confounding due to population stratification and cryptic relatedness in genomic data association analyses. Heredity (Edinb) 2011, 106, (4), 511-9.

3. The authors state that validating genetic studies with TCGA is limited because the genetic composition of the TCGA patients may be different that the patients in the independent study cohort. However, wouldn’t population substructure primarily impact the results from a GWAS if the significant SNPs varied in frequency between populations? 

That is completely accurate: population substructure does impact results from GWAS and should be accounted for. However, there have been increasing reports about how population background, or substructure, also affects gene expression patterns and clinical outcomes. For example, there are differences in gene expression patterns between women of European and African descent that are independent of other social and clinical variables and may confer poorer outcomes for patients from African descent (see new reference: Grunda  et al. BMC Res Notes 2012). In the Discussion section we also comment that similar trends have been noted in endometrial cancer, with Non-Hispanic Black (NHB) populations having a more aggressive, poorer outcome type of cancer not explained by clinical variables (see References 5 and 6).

We added a new sentence in the Introduction section to add this information about how population substructure influences expression analysis and clinical outcomes. Also we added a new reference supporting this concept: Grunda, J. M.; Steg, A. D.; He, Q.; Steciuk, M. R.; Byan-Parker, S.; Johnson, M. R.; Grizzle, W. E., Differential expression of breast cancer-associated genes between stage- and age-matched tumor specimens from African- and Caucasian-American Women diagnosed with breast cancer. BMC Res Notes 2012, 5, 248.

4. Have the authors considered how Fst values may be used in this context?

The reviewer makes an excellent suggestion. In the original manuscript, we did not consider FST statistics. We assessed population substructure independently in both UIHC and TCGA datasets without any assumptions about how many subpopulations could be present. Both STRUCTURE and ADMIXTURE can do that type of analysis. However, an FST analysis compares both populations with the same markers for both. As suggested by the reviewer, we performed FST analysis. The mean FST for our comparison between both UIHC and TCGA cohorts was 0.015. Considering that the potential range of FST statistics goes from 0 (for genetically similar populations) to 1 (for divergent populations), we interpreted that there were some differences between both populations. However, is difficult to make further inferences from these data. We have added these results to the Results section, described the method in the Material and Methods section (4.5 Data analysis), and added three new references: Wright, S., Evolution in Mendelian Populations. Genetics 1931, 16, (2), 97-159; Meirmans, P. G.; Hedrick, P. W., Assessing population structure: F(ST) and related measures. Mol Ecol Resour 2011, 11, (1), 5-18; Zheng, X.; Levine, D.; Shen, J.; Gogarten, S. M.; Laurie, C.; Weir, B. S., A high-performance computing toolset for relatedness and principal component analysis of SNP data. Bioinformatics 2012, 28, (24), 3326-8. We also added a detailed description of the results in Appendix B.

5. Why did the authors choose ADMIXTURE and STRUCTURE? Given this is presented as a methods paper, greater detail about these algorithms and how they differ from each other would be appreciated. Is there significance to the differences in optimal K subpopulation structure for TCGA patients detected between the two algorithms? 

As requested, we added a paragraph in the Material and Method section (4.5 Data analysis) explaining in greater detail the type of methods that STRUCTURE and ADMIXTURE are, the principals they follow, their differences, and why they were chosen. We also reference the original papers that described these two methods: Alexander, D. H.; November, J.; Lange, K., Fast model-based estimation of ancestry in unrelated individuals. Genome Res 2009, 19, (9), 1655-64; Pritchard, J. K.; Stephens, M.; Donnelly, P., Inference of population structure using multilocus genotype data. Genetics 2000, 155, (2), 945-59.

Regarding the differences between both methods and TCGA results, we added a few sentences in the last paragraph of the Discussion section where we commented on limitations of the study. Briefly, we noted that choosing the optimal number of clusters, or K, is a difficult statistical problem that is somehow subjective and dependent on the knowledge of the underlying population. Therefore, small differences may be due to interpretation of the results and may not be significant.

6. The central issue of accounting for population substructure in a validation study is relevant beyond TCGA. Why did the authors frame it as an issue specific to TCGA?

As we described in our response to the first question/comment, our initial motivation for the study was to assess why the prediction models for clinical outcomes were not validated in TCGA dataset.  That is the reason we specifically compared the population structure of our cohort with TCGA population.

TCGA is a widely used and excellent asset for researchers studying genomic features of diverse malignancies, and we thought that our analysis would be of interest to all these researchers.

7. Minor: the authors should consider changing the spelling of “publically” to the more commonly used “publicly”

It has been corrected.

Reviewer 2 Report

This manuscript concisely describes the subpopulation within a cohort study and a TCGA dataset and raised a concern related to that in using publicly available datasets for the validation of genomics studies. The results are clear and well discussed, although the Introduction could be giving more backgrounds.

However, this kind of analysis usually uses genomic DNA sequencing data, but this study has used RNA-seq data. Although RNA-seq data can be used for calling genomic variants, it is not designed for such analysis in the first place. TCGA should probably have whole genome sequencing data, or at least exome sequencing data, which are far more appropriate for variants calling. I suggest retrying the analysis using whole genome or whole exome sequencing data. If it is not available for UIHC, at least TCGA part should use DNA sequencing data.

Also, a bit of background of ADMIXTURE and STRUCTURE packages, such as what they do and what their result implies, etc, would be very helpful.

Author Response

Reviewer 2.

This manuscript concisely describes the subpopulation within a cohort study and a TCGA dataset and raised a concern related to that in using publicly available datasets for the validation of genomics studies. The results are clear and well discussed, although the Introduction could be giving more backgrounds.

1. However, this kind of analysis usually uses genomic DNA sequencing data, but this study has used RNA-seq data. Although RNA-seq data can be used for calling genomic variants, it is not designed for such analysis in the first place. TCGA should probably have whole genome sequencing data, or at least exome sequencing data, which are far more appropriate for variants calling. I suggest retrying the analysis using whole genome or whole exome sequencing data. If it is not available for UIHC, at least TCGA part should use DNA sequencing data.

We completely agree with the reviewer: for population substructure analysis ideally we would use genotypes from sequencing or genotype microarrays (available in TCGA). However, the motivation of this analysis was to understand why the validation of our prediction model with integration of molecular and clinical data did not validate completely in TCGA data. We unfortunately do not have DNA sequencing data from the UIHC cohort, so we tried to use a method/s that would help us to compare population substructure using the data that we did have. Also, we considered performing TCGA genotyping with the DNA arrays that are available. However, we thought that approach would introduce more possible bias/differences by using two different technologies, platforms, methods pipeline and two different biological materials: DNA and RNA. Therefore, we decided to use the results of RNA sequencing and extract genotypes from both cohorts of patients, UIHC and TCGA. We used exactly the same approach to minimize differences due to methodology: alignment, VCF file creation and genotype extraction, filtering with PLINK, imputation of missing genotypes, and substructure analysis with ADMIXTURE or STRUCTURE.

We added an initial paragraph at the beginning of the Materials and Methods section explaining why we used RNA-seq datasets to extract out genotypes for both cohorts of patients.

2. Also, a bit of background of ADMIXTURE and STRUCTURE packages, such as what they do and what their result implies, etc, would be very helpful.

As requested, we added a paragraph in the Material and Method section (4.5 Data analysis) explaining in greater detail the type of methods that STRUCTURE and ADMIXTURE are, the principals they follow, their differences, and why they were chosen. We also reference the original papers that described these two methods: Alexander, D. H.; November, J.; Lange, K., Fast model-based estimation of ancestry in unrelated individuals. Genome Res 2009, 19, (9), 1655-64; Pritchard, J. K.; Stephens, M.; Donnelly, P., Inference of population structure using multilocus genotype data. Genetics 2000, 155, (2), 945-59.

Round 2

Reviewer 1 Report

The authors addressed the concerns I had raised. No further comments.

Reviewer 2 Report

I still think that adding in the subpopulation analysis result from DNA sequencing data from TCGA would be better. However, given that this manuscript is more about presenting an example where TCGA dataset is inappropriate for validation, RNA-seq data might be just acceptable.

Other than that, the lead-up paragraph in the Materials and Methods section sounds more like Introduction or Discussion. Authors might need to consider moving them to a more suitable part.